# Identification and Validation of Key Biomarkers in the Proximal Aqueous Humor Outflow Pathway

**DOI:** 10.3390/cimb47030147

**Published:** 2025-02-25

**Authors:** Rong Du, Enzhi Yang, Madison Clark, Ningli Wang, Yiqin Du

**Affiliations:** 1Department of Ophthalmology, Morsani College of Medicine, University of South Florida, Tampa, FL 33612, USA; 2Beijing Institute of Ophthalmology, Beijing Tongren Eye Center, Beijing Tongren Hospital, Capital Medical University, Beijing Ophthalmology & Visual Science Key Laboratory, Beijing 100730, China

**Keywords:** glaucoma, trabecular meshwork, Schlemm’s canal, microarray, scRNA-seq, DEGs, biomarkers

## Abstract

Glaucoma is a leading cause of irreversible blindness, with elevated intraocular pressure (IOP) as the most important risk factor. The trabecular meshwork (TM) and Schlemm’s canal are the main components of the proximal aqueous humor outflow pathway. Their dysfunction is a major contributor to IOP elevation. This study aims to identify and validate key biomarkers for TM and Schlemm’s canal endothelial (SCE) cells. A Microarray was performed on characterized human TM and SCE cells to analyze their transcriptome profiling. Differentially expressed genes (DEGs) were identified and cross-referenced with published single-cell RNA sequencing (scRNA-Seq) datasets to ensure cell-specific relevance. Further validation was performed using qPCR and re-confirmed on the scRNA-seq datasets. One-way ANOVA was used for statistical analysis, and *p* < 0.05 was considered significant. The Microarray revealed 341 DEGs, with TM cells enriched in metabolic and signaling pathways and SCE cells enriched in adhesion, immune, and morphogenesis-related processes. Cross-referencing with scRNA-Seq data refined the list of candidate biomarkers, and qPCR confirmed the significant gene expression differences between TM and SCE cells. CTTNBP2 and MGARP were identified as TM cell markers. JAM2, PODXL, and IFI27 are new SCE cell biomarkers. The validated biomarkers offer insights into glaucoma pathophysiology and lay the groundwork for targeted therapies.

## 1. Introduction

Glaucoma is a leading cause of irreversible blindness worldwide, with the most common form being primary open-angle glaucoma (POAG) [1,2,3]. The global prevalence is 3.05% for POAG compared to 0.50% for primary angle closure glaucoma (PACG) for the population between 40 and 80 years old [1]. Glaucoma affects millions of people worldwide and is projected to reach 111.8 million in 2040 [1]. Elevated intraocular pressure (IOP), the most important risk factor of glaucoma, results from increased resistance of the aqueous outflow, mainly the conventional aqueous humor outflow pathway that accounts for up to 90% of total aqueous drainage [4], comprising the trabecular meshwork (TM) and Schlemm’s canal [5,6,7]. Dysfunction in this pathway leads to IOP elevation, optic nerve damage, and irreversible vision loss [8,9,10,11,12].

A major challenge in studying the pathophysiology of the outflow pathway in glaucoma is the lack of definitive biomarkers that can reliably distinguish TM cells and Schlemm’s canal endothelial (SCE) cells. Many of the currently identified markers, such as aquaporin-1 (AQP1), chitinase-3-like-1 (CHI3L1), and matrix Gla protein (MGP) [9,13,14,15,16,17], are not exclusive to TM cells and are also expressed in other tissues and cell types [18,19,20]. The only acceptable identification for TM cells is their responsiveness to dexamethasone with increased expression of myocilin (MYOC) [21]. SCE cells are highly specialized endothelial cells with dual characteristics of both blood and lymphatic vessels. This unique hybrid nature is reflected in the expression of markers typically associated with vascular endothelial cells (CD31 and VEGFR2) and lymphatic markers, such as VEGFR3, PROX1, and podoplanin (PDPN) [22,23,24,25,26]. However, as SCE cells mature through the process of canalogenesis, they gradually lose certain traits, making it even more challenging to pinpoint stable markers for their identification and track their function in glaucoma pathophysiology. Despite their critical role in aqueous humor outflow and IOP regulation, much remains unknown about their molecular identity and specific characteristics in normal and glaucomatous conditions. The current research struggles to fully differentiate SCE cells from surrounding TM cells due to the overlap in marker expression and the progressive changes that occur during development and in the disease status.

Recent advancements in gene expression technologies, particularly RNA sequencing, offer powerful tools to overcome these limitations. Microarray-based analysis provides an overview of gene expression across entire tissues, while single-cell RNA sequencing (scRNA-Seq) allows high-resolution analysis of individual cells, revealing cell-type-specific gene expression. This technology is particularly useful for identifying molecular differences between closely related cell types, such as TM cells and SCE cells. However, comprehensive studies utilizing both approaches in tandem to identify distinct biomarkers for these key cell populations are still lacking.

In this study, we aimed to identify novel biomarkers that can distinguish TM cells and SCE cells. Using our primary cultured TM and SCE cells for the Microarray assay and publicly available scRNA-Seq data [27] for double-validation, we analyzed the molecular differences between TM cells and SCE cells. Additionally, we performed qPCR on three TM cell strains and three SCE strains to validate the specific candidate biomarkers identified from both the Microarray and scRNA-Seq datasets.

## 2. Materials and Methods

### 2.1. Cell Culture and Sample Preparation

Human TM cells were isolated from de-identified donor corneal tissue and cultured as previously described [17,21]. Identification was performed by immunofluorescent staining and qPCR on the expression level changes of CHI3L1 and MYOC before and after dexamethasone treatment, as previously described [28,29,30]. Human SCE cells were also isolated from de-identified donor corneal tissue, cultured, and characterized based on the methods previously reported by Stamer et al. [31,32]. In brief, cells were isolated from the Schlemm’s canal lumen of human donor eyes using a cannulation technique. These cells were then identified by the positive expression of vascular endothelial cadherin (VE-CAD, the gene also known as *CDH5*), PROX1, and Fibulin-2 using immunofluorescent staining and qPCR. Both TM cells and SCE cells were cultured in Dulbecco’s Modified Eagle Medium/F12 (DMEM/F12) supplemented with 10% fetal bovine serum (FBS), 100 IU/mL of penicillin, and 100 μg/mL of streptomycin and maintained at 37 °C with 5% CO_2_. The cell culture media were changed every 3 days, and cells were passaged when they reached 95–100% confluency. Both TM and SCE cells were used between passages 3 and 5. Each experiment was repeated with at least three biological and technical replicates. Detailed donor information from which the cells were isolated and cultured is listed in Table 1.

### 2.2. RNA Isolation and Microarray Profiling Analysis

Cells were cultured and passaged as described above until reaching 95–100% confluency, followed by cell lysate and total RNA extraction using RLT and an RNeasy Kit (Qiagen, Valencia, CA, USA) [17,33]. RNA quality and concentration were assessed using a NanoDrop spectrophotometer (Thermo Fisher, Waltham, MA, USA). A Microarray assay was performed using the Affymetrix Human Genome U133 Plus 2.0 Array platform. Differentially expressed genes (DEGs) were identified using the RMA algorithm for normalization and the Limma package for statistical analysis. A false discovery rate (FDR) < 0.05 and log2 fold change > 1 were set as thresholds for significance. The results were analyzed using volcano plots to identify significantly upregulated and downregulated genes in each cell type. Gene Ontology (GO) enrichment analysis was performed to identify biological processes, cellular components, and molecular functions associated with the DEGs using the clusterProfiler package in *R* programming language. Additionally, Gene Set Enrichment Analysis (GSEA) was conducted to identify enriched Kyoto Encyclopedia of Genes and Genomes (KEGG) pathways, providing insights into the functional differences between TM cells and SCE cells. Pathways with an adjusted *p*-value < 0.05 were considered significant.

### 2.3. Single-Cell RNA Sequencing (scRNA-Seq) Validation

To validate the DEGs identified through microarray analysis, we utilized a publicly available scRNA-Seq dataset [27]. First, cells from the aqueous humor outflow pathway were identified and classified into distinct subpopulations using established biomarkers from the published literature [27]. Differential gene expression analysis was performed to compare these subpopulations, specifically focusing on TM cells and SCE cells. Venn plots were generated to identify overlapping DEGs between our Microarray dataset and the scRNA-Seq dataset. Furthermore, uniform manifold approximation and projection (UMAP) was utilized to visualize the spatial distribution and expression levels of the overlapping DEGs across the cell populations.

### 2.4. Quantitative Reverse Transcription–Polymerase Chain Reaction (qPCR)

Cells were lysed with RLT buffer, and RNAs were isolated using an RNeasy Kit, as mentioned above. cDNAs were synthesized from the RNAs using reverse transcriptase (SuperScript III; Invitrogen, Waltham, MA, USA). qPCR was conducted using SYBR Green dye (Thermo Fisher, Pittsburgh, PA, USA). Primers for target genes were either newly designed using the NIH Primer-BLAST tool or previously used and published, with references cited in Table 2 of where the sequences were provided. The amplification of 18S rRNA was used as a normalization control, and negative controls without cDNA were included in each assay. Relative mRNA abundance was calculated using the ^ΔΔ^Ct method as previously described [17,34,35]. Three independent biological replicates from three different donors of cells and isolated cDNAs were analyzed, with each reaction performed in triplicate as technical repeats.

### 2.5. Immunofluorescent and Hematoxylin–Eosin (H&E) Staining

For immunofluorescent staining, cells were fixed in 4% paraformaldehyde for 15 min, permeabilized with 0.5% Triton X-100, and blocked with 1% bovine serum albumin (BSA) for 1 h at room temperature. Cells were incubated with the primary antibodies MYOC (a generous gift from Dr. Stamer at Duke University), CHI3L1 [28], PROX1, Fibulin 2 (R&D Systems), and VE-CAD (Abcam, Waltham, MA, USA) overnight at 4 °C. After three washes with PBS, the corresponding fluorescent secondary antibodies and 4′,6-diamidino-2-phenylindole (DAPI, Thermo Fisher) were applied to the sections for 2 hrs. After five washes, the slides were mounted, imaged, and analyzed using a fluorescence microscope (Keyence BZ-X800, Itasca, IL, USA).

For H&E staining, tissues were fixed in 10% formalin, embedded in paraffin, and sectioned at 8 µm. Sections were deparaffinized, rehydrated, stained with hematoxylin for nuclei (blue-purple), counterstained with eosin for cytoplasm (pink), dehydrated, cleared in xylene, and mounted for histological analysis. The slides were imaged and analyzed using a Keyence microscope.

### 2.6. Data Analysis

The statistical differences were analyzed by one-way or two-way ANOVA followed by Tukey’s multiple comparisons test using GraphPad Prism version 10.2.2. *p* < 0.05 was considered statistically significant. All data analyses were conducted impartially and unbiased.

## 3. Results

### 3.1. Validation of TM and SCE Cell Identity

Cultured human TM cells and SCE cells between passages 3 and 5 underwent immunofluorescent staining and qPCR to confirm the distinct identities of TM and SCE cells (Figure 1). Using a cannulation technique, primary SCE cells were cultured on a 6-0 nylon suture ex vivo, with cell nuclei staining indicating the cells grown on the suture as passage 0 (Figure 1A–C). Passaged SCE cells were detected with the positive staining of Fibulin-2, VE-CAD, and PROX1 (Figure 1D,E), which are currently acceptable SCE markers. Figure 1F,G show the human TM tissue location on the cornea. Figure 1H shows hematoxylin and eosin staining on a human corneal section, indicating the TM and Schlemm’s canal structure and location. Figure 1I,J are the representative staining results of the differential expression of CHI3L1 and MYOC in cultured TM cells (I) and TM cells treated with 100 nM dexamethasone for 7 days (J). After dexamethasone treatment, TM cells had reduced expression of CHI3L1 (green) and increased expression of MYOC (red). Figure 1K,L are the qPCR results showing dexamethasone-treated TM cells significantly reduced *CHI3L1* expression (9.33-fold lower) and increased *MYOC* expression (22.65-fold higher) compared to untreated TM cell controls. SCE cells have significantly increased expression of *VE-CAD* (12.85-fold higher), *Fibulin 2* (1315.45-fold higher), and *PROX-1* (10.85-fold higher) compared to TM cells (Figure 1M–O). These findings confirm the successful culture and identification of human TM and SCE cells used in this study.

### 3.2. Microarray Assay Identifying Differentially Expressed Genes in TM and SCE Cells

Microarray analysis reveals a total of 341 DEGs between TM cells and SCE cells, with 164 upregulated (red dots in Figure 2A) and 177 downregulated genes (blue dots in Figure 2A) in TM cells compared to SCE cells. Figure 2A is the volcano plot of DEGs, highlighting significant upregulated and downregulated genes in TM cells and SCE cells.

Functional enrichment analysis of the DEGs was performed to understand their biological significance. GO enrichment analysis reveals that TMC-enriched DEGs (Figure 2B) were predominantly enriched in the cGMP biosynthetic process, nitric oxide-mediated signal transduction, and cholesterol biosynthetic and metabolic processes. SCE-enriched DEGs (Figure 2C) were associated with distinct processes encompassing molecular interactions, development, and morphogenesis; physiological behavior; cell proliferation; gap junction assembly regulation; and vitamin D receptor signaling.

The GSEA of KEGG pathways highlights distinct functional themes for TM and SCE cells. For TM cells (Figure 2D), enrichment pathways were observed in the cell cycle and replication, fundamental metabolism, biosynthesis, and transport processes. In contrast, SCE cells (Figure 2E) showed significant enrichment in cytoskeletal dynamics and adhesion, immune and inflammatory signaling, and osteoclast and lysosome pathways. Figure 2F presents a heatmap illustrating the differential expression profiles of TM-enriched and SCE-enriched DEGs. These patterns underscore the distinct metabolic, proliferative, and signaling pathways that define TM and SCE cell functions.

### 3.3. Validation with scRNA-Seq Data

We utilized publicly available scRNA-Seq datasets to identify 11 clusters in aqueous humor outflow for clustering analysis, and cell homology was found using cluster-specific genes and typical cell type markers (Figure 3A,B). The cell types include TM cells, SCE cells, pericytes, myelinating Schwann (Schmy) cells, nonmyelinating Schwann (SchNmy) cells, ciliary cells, mast cells, macrophages, T/NK cells, B cells, and melanocytes. Differential expression analysis was performed to generate volcano plots for these cell populations (Figure 3C,D). Subsequently, we performed an intersection analysis between the scRNA-Seq results and our Microarray data, focusing on TM and SCE separately (Figure 3E,F). This analysis reveals eight overlapping genes for TM cells and eight for SCE cells.

To further validate these findings, we projected the 16 intersecting genes (Figure 3E,F) onto a UMAP plot to determine their expression across other cell populations within the aqueous humor outflow pathway (Figure 4). Notably, we identified CTTNBP2 and MGARP as TM-specific genes (Figure 4A), while JAM2, IFI27, PODXL, and LRRC32 were found to be relatively SCE-specific (Figure 4B). These findings highlight the spatial and cell-specific expression patterns of the overlapping DEGs, further supporting their biological relevance.

### 3.4. qPCR Validation of Key DEGs

qPCR was used to validate the selected DEGs from the intersected Microarray and scRNA-Seq datasets using cultured human TM and SCE cells identified by the same criteria, as shown in Figure 2. Figure 5A shows the qPCR results, indicating fold-change differences for each gene between TM cells and SCE cells. *CTTNBP2* and *MGARP* exhibited significantly higher expression levels in TM cells, consistent with both the Microarray and scRNA-Seq data, with average fold changes of 37.16 and 25.05, respectively, compared to SCE cells. Similarly, *JAM2*, *PODXL*, and *IFI27* were validated as specific genes for SCE cells, with average fold changes of 5.52, 95.86, and 143.95, respectively. However, *LRRC32*, initially hypothesized to be enriched in SCE cells, did not show significant expression differences between TM cells and SCE cells.

To further confirm the identified genes and commonly used cell markers we mentioned in the Introduction, we verified the gene expression profile in the scRNA-seq data. The violin plots in Figure 5B demonstrate that the newly identified markers *CTTNBP2*, *MGARP*, *JAM2*, *IFI27*, and *PODXL* exhibited higher and more distinct expression patterns in TM or SCE cell clusters. In contrast, the expression levels of traditional markers, such as *CHI3L1*, *MYOC*, *AQP1*, *MGP*, *CRYAB*, *PROX1*, and *FBLN2* (Figure 5C), were high or low expressed across most cell clusters without distinct differences, indicating limited specificity and sensitivity in distinguishing distinct cell populations.

Table 3 provides an overview of the validated genes, including their full names and primary functions (https://www.genecards.org/ accessed on 20 January 2025). These findings reinforce the reliability of our multi-platform approach while refining the list of candidate genes for further functional studies.

## 4. Discussion

In this study, we successfully cultured and identified human TM cells and SCE cells based on TM cell responsiveness to dexamethasone treatment [21,28,30] and SCE cell expression of PROX-1, Fibulin 2, and VE-CAD [11,13,24,25]. Further, we performed a Microarray assay on our identified cells to discover the distinct gene expression profiling and pathways of TM and SCE cells. We compared our Microarray data with published scRNA-Seq data [27], identified specific genes of TM cells and SCE cells, and validated our discovered genes by qPCR. The differential expression patterns in TM cells and SCE cells we identified in this study align well with their distinct physiological roles in the aqueous humor outflow pathway. To our knowledge, this is the first report identifying molecular-level differences between human TM cells and SCE cells.

TM cells are structurally adapted to regulate resistance within the outflow pathway. Notably, the upregulation of *CTTNBP2* and *MGARP* in TM cells suggests a critical role in maintaining cytoskeletal integrity and mitochondrial energy production, both of which are essential for mechanical load adaptation [37]. Biomechanics of the TM is an important factor of glaucoma [38], and increased TM stiffness is one of the biomechanical properties of glaucoma with increased IOP [39,40]. This could be a new target for developing novel therapies for glaucoma focusing on TM pathophysiology. *CTTNBP2* has been identified as being associated with glaucoma through GWASs [41], and our research reveals its highly specific expression in TM cells, potentially linking it to microtubule dynamics, which may adapt to dynamic aqueous humor outflow and fluctuating IOP. *MGARP*, involved in mitochondrial function and oxidative stress response [42], may reflect TM cells’ high metabolic and antioxidant demands [43]. Its dysregulation could disrupt mitochondrial homeostasis, accelerating TM cell damage under oxidative stress, a key feature in glaucoma [44,45].

Functional enrichment analysis provides deeper insights into the specialized roles of TM cells, emphasizing their dynamic metabolic and biosynthetic activities essential for maintaining structural and functional integrity under varying intraocular pressures. GO analysis reveals significant enrichment in biosynthetic and signal transduction processes, such as the cGMP biosynthetic process and nitric oxide-mediated signal transduction, underscoring the importance of nitric oxide signaling in regulating TM relaxation and enhancing aqueous humor outflow, which is consistent with the published reports [46,47]. Additionally, enrichment in cholesterol biosynthetic and metabolic processes highlighted active lipid metabolism, which is crucial for maintaining cell membrane integrity and intracellular signaling. Processes like Ras protein signal transduction and sterol biosynthesis further reflected the TM’s capacity to sustain energy demands and adapt to dynamic pressure changes. The complementary KEGG pathway analysis supports these findings, showing pathways related to cell cycle and DNA replication, indicative of active cellular turnover and repair mechanisms. Enrichment in steroid biosynthesis, purine metabolism, nitrogen metabolism, and terpenoid backbone biosynthesis emphasized the TM’s high metabolic demands for modulating aqueous humor dynamics. Moreover, pathways like folate transport and butanoate metabolism highlighted roles in redox balance and cellular signaling. Collectively, these findings illustrate the TM’s specialized metabolic and biosynthetic functions, which are critical for maintaining homeostasis and responding to the stress of intraocular pressure fluctuations.

In contrast, the enriched pathways in SCE cells reflect their endothelial nature and involvement in immune response and vascular regulation. The upregulation of *JAM2*, *PODXL*, *and IFI27* in SCE cells highlights their contributions to maintaining barrier integrity, endothelial polarity, and immune surveillance. *JAM2* supports endothelial barrier integrity and regulates immune cell trafficking [48], while *PODXL* preserves the glycocalyx, which is expressed on the vascular endothelium and is essential for maintaining aqueous humor outflow [49,50]. This signifies a novel discovery of an SCE cell marker representing SCE cell vascular characteristics. *IFI27*, associated with oxidative stress and inflammation responses, reflects SCE cells’ role in mitigating environmental stressors. The *IFI27* gene has been detected to be associated with IOP elevation and POAG [51].

The SCE cells demonstrated enrichment in biological processes related to cell adhesion, immune signaling, and cytoskeletal regulation, emphasizing their critical roles in maintaining endothelial barrier function and immune modulation. The GO analysis highlights key processes, such as delta-catenin binding, which underscores the importance of stabilizing endothelial cell junctions essential for barrier integrity. Enrichment in mesenchymal to epithelial transition suggested cellular plasticity, enabling adaptive responses during injury or repair. The regulation of gap junction assembly highlights the importance of intercellular communication in sustaining endothelial function, while the vitamin D receptor signaling pathway points to systemic regulatory influences on SCE cell activity. KEGG pathway analysis further supports these findings, with enrichment in cytokine–cytokine receptor interactions and the NF-kappa B signaling pathway, reflecting active roles in immune and inflammatory responses that are involved in IOP regulation [52]. Pathways such as the regulation of actin cytoskeleton and cell adhesion molecules emphasize SCE cells’ specialization in preserving barrier stability. Enrichment in IL-17 and TNF signaling pathways potentially suggests the involvement of SCE cells in immune modulation and inflammation, while lysosome activity and leukocyte transendothelial migration highlight SCE cells’ roles in cellular trafficking and turnover [53]. Collectively, these findings demonstrate the SCE cells’ unique contributions to maintaining a functional endothelial barrier and regulating immune processes, both of which are vital for ensuring unobstructed aqueous humor outflow.

While this study provides significant insights with ample biological and technical repeats and cross-referencing with published scRNA-seq datasets, several limitations should be acknowledged. First, the reliance on in vitro primary cultures may not fully capture the complex in vivo microenvironment of the TM and Schlemm’s canal. Future studies utilizing advanced organoid models or in vivo systems are warranted to address these limitations. Second, our analyses did not differentiate the three distinct TM subpopulations or the inner and outer walls of the Schlemm’s canal, potentially overlooking crucial functional nuances. Furthermore, the functional roles of the identified DEGs, particularly those of IFI27 and MGARP, remain to be elucidated. Functional assays and gene knockout studies could provide deeper insights into their contributions to TM and SC physiology and glaucoma pathogenesis.

## 5. Conclusions

This study provides a comprehensive transcriptomic analysis of TM and SCE cells, highlighting their distinct molecular roles in aqueous humor outflow regulation. By integrating Microarray, scRNA-Seq, and qPCR analyses, we identified key biomarkers, including *CTTNBP2*, *MGARP*, *JAM2*, *PODXL*, and *IFI27*, that reflect the specialized functions of these cell types. These findings advance our understanding of glaucoma pathophysiology and establish a foundation for future therapeutic strategies targeting TM and SC dysfunction. Developing new therapies that focus on regulating these genes and related pathways could offer a promising approach to effectively treat glaucoma with a one-time treatment, eliminating the need for daily eye drops or repeated surgeries.

## Figures and Tables

**Figure 1 cimb-47-00147-f001:**
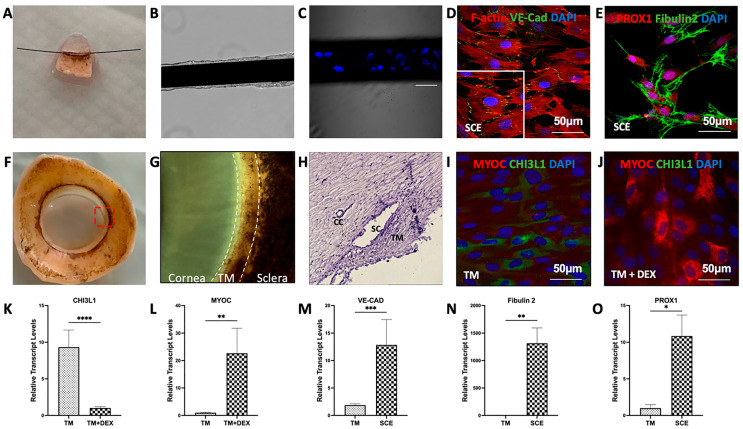
Isolation, culture, and identification of trabecular meshwork cells and Schlemm’s canal endothelial cells. (**A**–**C**) Images indicating the suture cannulation of SCE cell culture. (**A**) shows a bright-field image of a 6-0 nylon suture passing through the Schlemm’s canal. (**B**) depicts cells growing on the suture surface, and (**C**) displays DAPI-stained nuclei of cells grown on the suture. (**D**) shows passaged SCE cells stained with F-actin in red, VE–cadherin (VE-CAD) in green, and DAPI in blue for nuclei counterstaining, providing insights into the cytoskeletal structure and cell–cell junctions. (**E**) highlights PROX1 in red, Fibulin-2 in green, and DAPI in blue, demonstrating positive staining of PROX1 and Fibulin-2 in SCE cells. (**F**,**G**) Representative images of human TM tissue captured under an inverted microscope, showing the structural characteristics of the TM. (**G**) is the magnified picture of the structure in the red frame in (**F**). (**H**) Hematoxylin and eosin (H&E) staining on a corneal section indicates the location of TM and Schlemm’s canal (SC) tissues, highlighting the cellular and tissue architecture of the region. Immunofluorescent staining of cultured TM cells (**I**) and dexamethasone-treated TM cells (**J**) with differential expression of CHI3L1 (green) and MYOC (red). Nuclei are counterstained with DAPI (blue). (**K**–**O**) qPCR analysis of differential gene expression on TM cells, dexamethasone-treated TM cells (TM-Dex), and SCE cells, detecting *CHI3L1*, *MYOC*, *VE-CAD*, *Fibulin-2*, and *PROX1* gene expression. Each graph represents mean ± SD of three biological replicates. * *p* < 0.05, ** *p* < 0.01, *** *p* < 0.001, and **** *p* < 0.0001. Scale bar: 50 μm.

**Figure 2 cimb-47-00147-f002:**
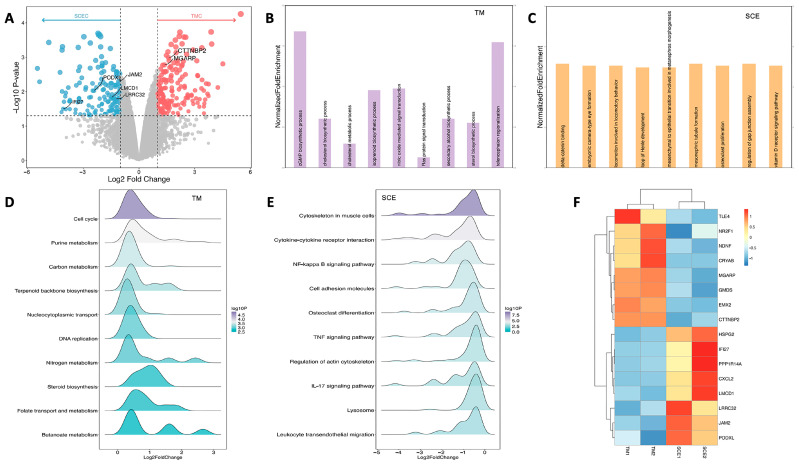
Transcriptomic analysis of TM and SCE cells. (**A**) Volcano plot shows differentially expressed genes (DEGs) between trabecular meshwork (TM) and Schlemm’s canal endothelial (SCE) cells. Gene Ontology (GO) enrichment analysis of TM-enriched DEGs (**B**) and SCE−enriched DEGs (**C**). Gene Set Enrichment Analysis (GSEA) of KEGG pathways enriched in TM cells (**D**) and SCE cells (**E**). (**F**) Heatmap of core DEGs distinguishing TM and SCE cells, illustrating relative expression levels across samples.

**Figure 3 cimb-47-00147-f003:**
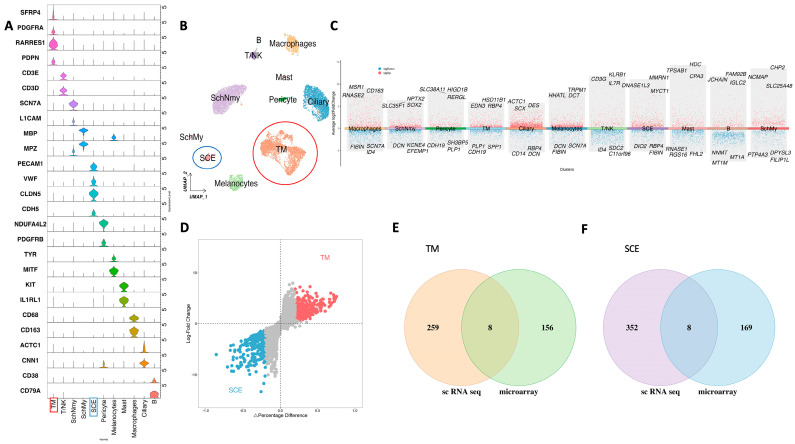
Identification and analysis of key genes in the aqueous humor outflow pathway. (**A**) Violin plot of cluster-specific biomarkers distinguishing the identified cell populations. Red box highlights TM cell population and blue box highlights SCE cell population. (**B**) UMAP plot showing the clustering of cells within the aqueous humor outflow pathway, highlighting distinct TM (red circle) and SCE (blue circle) cell populations. (**C**) Volcano plots illustrating the differential gene expression profiles across all clusters, identifying significant biomarkers. (**D**) Volcano plot of differential gene expression specific to trabecular meshwork (TM) and Schlemm‘s canal endothelial (SCE) cell clusters. Venn diagram displaying the intersecting genes between TM-specific (**E**) and SCE-specific (**F**) results from Microarray and scRNA-Seq analyses, identifying eight overlapping genes in TM and SCE cells, respectively.

**Figure 4 cimb-47-00147-f004:**
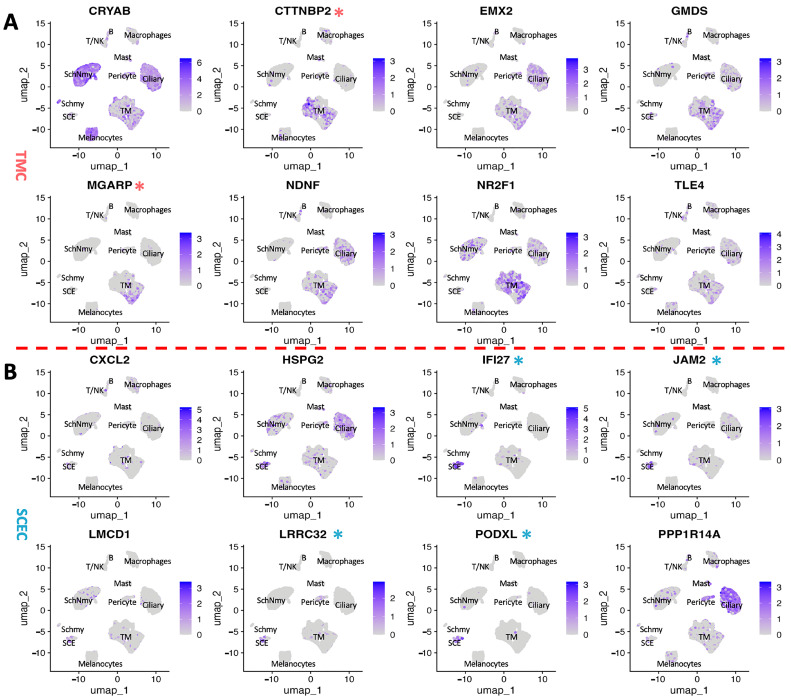
Validation of intersection genes using UMAP. UMAP visualization of intersection genes, highlighting their expression patterns across different cell clusters. (**A**) Red * marks the genes specifically expressed in TM cell cluster. (**B**) Blue * marks the genes specifically expressed in SCE cell cluster.

**Figure 5 cimb-47-00147-f005:**
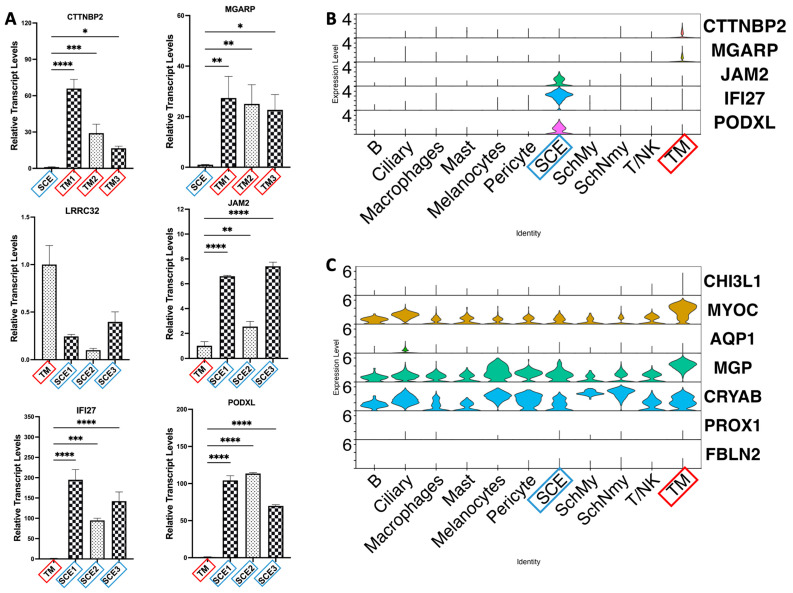
Validation and visualization of intersection genes using qPCR and violin plots. (**A**) qPCR validation for six differentially expressed genes, showing their relative expression levels in the experimental cell groups. (**B**) Violin plots showing identified genes in this study were explicitly expressed in the distinct TM or SCE cells. (**C**) Violin plots showing traditionally used gene markers were expressed high or low in most of the cell clusters, as identified in the scRNA-seq datasets. Red box highlights TM cell population and blue box highlights SCE cell population. Each graph represents mean ± SD of three technical replicates. * *p* < 0.05, ** *p* < 0.01, *** *p* < 0.001, and **** *p*< 0.0001.

**Table 1 cimb-47-00147-t001:** Donor information.

	Cell Type	Age	Gender	Source	Ethnicity
qPCR	TM1	55	Female	Human	Caucasian
	TM2	62	Male	Human	Caucasian
	TM3	50	Female	Human	African American
	SCE1	58	Male	Human	Caucasian
	SCE2	70	Female	Human	African American
	SCE3	45	Male	Human	Caucasian
Microarray	TM4	45	Female	Human	Caucasian
	TM5	54	Male	Human	Caucasian
	SCE4	27	Female	Human	African American
	SCE5	54	Male	Human	Caucasian

**Table 2 cimb-47-00147-t002:** Primer sequences used for qPCR in the study.

Gene Name	DNA Sequence
18S rRNA [17,28,34,36]	Forward: CCCTGTAATTGGAATGAGTCCAC
	Reverse: GCTGGAATTACCGCGGCT
VE-CAD (CDH5)	Forward: ATGACGTGAACGACAACTGG
	Reverse: TACATGACAGAGGCGTGGTC
PROX1	Forward: CAGCGGTCTCTCTAGTACAG
	Reverse: GCCTGCCAAAAGGGGAAAGA
FIBULIN 2	Forward: CCACTGCTACAAGGCACTCA
	Reverse: GCAGTAGAAGGAGCCCTTGG
CHI3L1 [17,29]	Forward: CCTTGACCGCTTCCTCTGTA
	Reverse: GTGTTGAGCATGCCGTAGAG
MYOC [17,36]	Forward: AAGCCCACCTACCCCTACAC
	Reverse: TCCAGTGGCCTAGGCAGTAT
CTTNBP2	Forward: CCCTCTCCATCCTTGAAGCAGT
	Reverse: GAAGCTTCTCCATTTCCAGCTTCT
MGARP	Forward: GTCACAGTCAGTGCTGGTGG
	Reverse: CAGTTTCCGCAACATTCTCCT
JAM2	Forward: GGATATCGCAGGTGTCCTGG
	Reverse: AAGGCCACAACTACTACGGC
PODXL	Forward: AGAAGCAGCTCGTCCTGAAC
	Reverse: ACTTATCTTGGGCCGGGTTG
LRRC32	Forward: CTTGACAGGAACGTCCGCAG
	Reverse: ACAGGGCACTTTGTCTTGGT
IFI27	Forward: CGGTGAGGTCAGCTTCACAT
	Reverse: GGCCACAACTCCTCCAATCA

**Table 3 cimb-47-00147-t003:** Gene symbols, full names, and functional roles of key genes.

	Gene Symbol	Full Name	Function
TM	CTTNBP2	Cortactin-Binding Protein 2	This gene encodes a protein with six ankyrin repeats and several proline-rich regions that may interact with a central regulator of the actin cytoskeleton.
	MGARP	Mitochondria Localized Glutamic Acid-Rich Protein	This gene plays a role in the trafficking of mitochondria along microtubules that are involved in axonal transport; cellular response to hormone stimulus; and protein targeting to mitochondrion.
SCE	JAM2	Junctional Adhesion Molecule 2	This gene encodes a type I membrane protein that is localized at the tight junctions of both epithelial and endothelial cells.
	PODXL	Podocalyxin-Like	This gene encodes a member of the sialomucin protein family that can bind in a membrane protein complex with a Na+/H+ exchanger regulatory factor to intracellular cytoskeletal elements and be expressed in vascular endothelium cells and bind to L-selectin.
	IFI27	Interferon Alpha Inducible Protein 27	This gene is a probable adapter protein involved in different biological processes.

## Data Availability

The data presented in this study are available upon request from the corresponding author Y.D.

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
