# Peer review of "Identification and Validation of Key Biomarkers in the Proximal Aqueous Humor Outflow Pathway"

_cimb, 2025, doi:10.3390/cimb47030147_

Round 1

Reviewer 1 Report

Comments and Suggestions for Authors

Very interesting study, well done. Just one comment: Could you discuss the clinical implications of your findings of these key markers? Have other studies looked into the clinical implications?

Author Response

Comments: Very interesting study, well done. Just one comment: Could you discuss the clinical implications of your findings of these key markers? Have other studies looked into the clinical implications?

Response: Thanks to the reviewer for the positive comments and for pointing this out. We agree with this comment, and we added the discussion in the manuscript as red on page 11, lines 294-297 as red.

Biomechanics of the TM is an important factor of glaucoma37 and increased TM stiffness is one of the biomechanical properties of glaucoma with increased IOP38,39. This could be a new target for developing novel therapies for glaucoma focusing on the TM pathophysiology.

We also added “IFI27 gene has been detected to be associated with IOP elevation and POAG50” on page 12, lines 354-355 as red.

Our report on these genes is the first one, and no other studies have looked into the clinical implications yet.

Reviewer 2 Report

Comments and Suggestions for Authors

I appreciate the effort of the Authors in this study, which is interesting and well-conducted. The Methodology is correct and the Results are presented clearly. However, the manuscript needs revision.

  1. The prevalence of glaucoma should be added in the Introduction section
  2. What is the confidence level of the study according to the Authors?
  3. The risk of bias should be evaluated 
  4. The clinical value of the study should be more highlighted in the Conclusion section

Author Response

Comments: I appreciate the effort of the Authors in this study, which is interesting and well-conducted. The Methodology is correct, and the Results are presented clearly. However, the manuscript needs revision.

Response: Thanks to the reviewer for the positive comments. We have revised the manuscript accordingly. Please see below.

Q1: The prevalence of glaucoma should be added in the Introduction section

Response: Thank you for pointing this out. We agree with this comment. Therefore, we have added the prevalence and population with reference cited in the Introduction marked as red. It is on page 1, lines 33-36.

The global prevalence is 3.05% for POAG compared to 0.50% for primary angle close glaucoma (PACG) for the population between 40 and 80 years old1. Glaucoma affects millions of people worldwide and is projected to 111.8 million in 20401.

Q2: What is the confidence level of the study according to the Authors?

Response: We are very confident of this study and the results, since we validated our own Microarray data with published scRNAseq data and using ample biological repeats and technical repeats to qPCR for our own data, and we validated our finding of the specific candidate biomarkers . We added a confident statement on page 12, lines 354-355 as red.

While this study provides significant insights with ample biological and technical repeats and cross-referencing with published scRNA-seq datasets, several limitations should be acknowledged.

Q3: The risk of bias should be evaluated.

Response: There is no bias issue for this study. For cultured human cells, we used cells from both male and female donors at a wide range of ages which are listed in Table 1. For data analysis, both our Microarray data and the published data are/will be published in public domains and data analysis was using R that everyone can use the data and check the accuracy.

Q4: The clinical value of the study should be more highlighted in the Conclusion section.

Response: We added the statement in Conclusion on page 13, line 372-374 as red.

Developing new therapies that focus on regulating these genes and related pathways could offer a promising approach to effectively treat glaucoma with a one-time treatment, eliminating the need for daily eyedrops or repeated surgeries.

To respond to this reviewer’s general evaluation on “Introduction and include relevant references”, we added new references numbered as #2, 3, 5, 6, 9, 10, 11, 28,  37, 38, 39, 44, 50 in the Introduction and throughout the whole manuscript.

Reviewer 3 Report

Comments and Suggestions for Authors Using Microarray methods and validation test, CTTNBP2 and MGARP were identified as TM cell markers. JAM2, PODXL and IFI27 are new SCE cell biomarkers.  The validated biomarkers offer insights into glaucoma pathophysiology and lay the groundwork for targeted therapies.    The research is well designed, pictures are of high quality and the manuscript is well written. As CTTNBP2 and MGARP were identified as TM cell markers. JAM2, PODXL and IFI27, it would be even clear for their distribution in TM or SCE cells is IM test were carried out.

Author Response

Comments: Using Microarray methods and validation test, CTTNBP2 and MGARP were identified as TM cell markers. JAM2, PODXL and IFI27 are new SCE cell biomarkers.  The validated biomarkers offer insights into glaucoma pathophysiology and lay the groundwork for targeted therapies.    The research is well designed, pictures are of high quality and the manuscript is well written. As CTTNBP2 and MGARP were identified as TM cell markers. JAM2, PODXL and IFI27, it would be even clear for their distribution in TM or SCE cells is IM test were carried out.

Response: The authors appreciate the reviewer’s positive comments and suggestions. We believe our results will have positive effects in the field and hopefully more novel therapies for glaucoma will be developed targeting glaucoma pathophysiology.

Round 2

Reviewer 2 Report

Comments and Suggestions for Authors

The manuscript has been revised insufficiently. Performance and reporting risk of bias should be described.

It is also worth adding the percentages of aqueous humor outflow pathway vie trabecular meshwork and Schlemm's canal.

Author Response

Comment 1: The manuscript has been revised insufficiently. Performance and reporting risk of bias should be described.

Response: Thanks to the reviewer for the suggestion. We added the statement on page 5, line 150 in red:

All data analyses were conducted impartially and unbiased.

Comment 2: It is also worth adding the percentages of aqueous humor outflow pathway vie trabecular meshwork and Schlemm's canal.

Response: Thanks to the reviewer for the suggestion. We added the statement with a new reference #4 on page 1, lines 37-38 as red.

Elevated intraocular pressure (IOP), the most important risk factor of glaucoma, results from increased resistance of the aqueous outflow, mainly the conventional aqueous humor outflow pathway that accounts for up to 90% of total aqueous drainage4